# Targeting TGF-β-Mediated SMAD Signaling Pathway via Novel Recombinant Cytotoxin II: A Potent Protein from *Naja naja oxiana* Venom in Melanoma

**DOI:** 10.3390/molecules25215148

**Published:** 2020-11-05

**Authors:** Afshin Derakhshani, Nicola Silvestris, Nima Hemmat, Zahra Asadzadeh, Mahdi Abdoli Shadbad, Niloufar Sadat Nourbakhsh, Leila Mobasheri, Parviz Vahedi, Morteza Shahmirzaie, Oronzo Brunetti, Hossein Safarpour, Behzad Baradaran

**Affiliations:** 1Immunology Research Center, Tabriz University of Medical Sciences, Tabriz 51656-65811, Iran; afshin.derakhshani94@gmail.com (A.D.); nima.hemmat1995@gmail.com (N.H.); zahraasadzadeh2834@gmail.com (Z.A.); abdoli.med99@gmail.com (M.A.S.); 2IRCCS Istituto Tumori “Giovanni Paolo II” of Bari, 70124 Bari, Italy; n.silvestris@oncologico.bari.it (N.S.); dr.oronzo.brunetti@tiscali.it (O.B.); 3Department of Biomedical Sciences and Human Oncology (DIMO), University of Bari, 70124 Bari, Italy; 4Department of Biology, Islamic Azad University, Varamin-Pishva Branch, Varamin, Tehran 33817-74895, Iran; Niloufarnourbakhsh1994@gmail.com; 5Department of Immunology, Faculty of Medicine, Birjand University of Medical Sciences, Birjand 97178-53577, Iran; mobasheri.l91@gmail.com; 6Department of Anatomical Sciences, Maragheh University of Medical Sciences, Maragheh 51656-65931, Iran; pa.vahedi48@gmail.com; 7Pharmaceutical Sciences Research Center, Shahid Beheshti University of Medical Sciences, Niayesh Highway, Valiasr Ave, Tehran 19919-53381, Iran; m.shahmirzaie@yahoo.com; 8Cellular & Molecular Research Center, Birjand University of Medical Sciences, Birjand 97178-53577, Iran; 9Department of Immunology, Faculty of Medicine, Tabriz University of Medical Sciences, Tabriz 51666-14766, Iran

**Keywords:** melanoma, snake venom, recombinant cytotoxin-II, TGF-β pathway, apoptosis

## Abstract

Since the current treatments have not resulted in the desired outcomes for melanoma patients, there is a need to identify more effective medications. Together with other snake venom proteins, cytotoxin-II has shown promising results in tumoral cells. In this study, recombinant cytotoxin-II (rCTII) was expressed in SHuffle^®^ T7 Express cells, while the epitope mapping of rCTII was performed to reveal the antibody-binding regions of rCTII. The MTT (3-(4,5-dimethylthiazol-2-yl)-2,5-diphenyltetrazolium bromide) assay was used to assess the viability of SK-MEL-3 and HFF-2 cells after treating these cells with rCTII. The qRT-PCR was performed to evaluate the expression levels of matrix metallopeptidase 3 (*MMP-3*), *SMAD2*, *SMAD3*, caspase-8, caspase-9, and miR-214 in order to reveal the rCTII-induced signaling pathways in melanoma. Our results have shown that two regions of amino acids, 6–16 and 19–44, as predicted epitopes of this toxin, are essential for understanding the toxicity of rCTII. Treating the melanoma cells with rCTII substantially inhibited the transforming growth factor-beta (TGF-β)–SMAD signaling pathway and down-regulated the expression of *MMP-3* and miR-214 as well. This cytotoxin also restored apoptosis mainly via the intrinsic pathway. The down-regulation of *MMP-3* and miR-214 might be associated with the anti-metastatic property of rCTII in melanoma. The inhibitory effect of rCTII on the TGF-β signaling pathway might be associated with increased apoptosis and decreased cancer cell proliferation. It is interesting to see that the IC50 value of rCTII has been lower in the melanoma cells than non-tumoral cells, which may indicate its potential effects as a drug. In conclusion, rCTII, as a novel medication, might serve as a potent and efficient anticancer drug in melanoma.

## 1. Introduction

Due to its significant incidence and mortality, melanoma has remained one of the troublesome malignancies, especially among the fair-skinned populations [1]. As a matter of fact, melanoma is responsible for most skin cancer-associated mortalities [2]. Moreover, the available therapies have not resulted in the desired outcomes in affected patients [3,4].

Multiple animal-derived secretions, e.g., snake venom toxins, have shown promising results in tumoral cells [5,6]. The cytotoxicity of snake venoms has been associated with stimulating apoptosis and inhibiting the migration of tumoral cells [7,8]. Cytotoxin I and cytotoxin II are two cytotoxins from *Naja naja oxiana* (Caspian cobra), which have been well-known for their cytotoxic effect on tumoral cells [9]. Since cytotoxin-II can substantially stimulate the intrinsic apoptosis pathway of tumoral cells, it can exert remarkable cytotoxicity in breast cancer with minimal effects on non-tumoral cells [10].

The transforming growth factor-beta (TGF-β) signaling pathway plays a critical role in cell growth, differentiation, apoptosis, migration, and cancer development [11]. Following its binding to the protein receptor, the intracellular SMAD family induces the gene expression. Since the TGF-β signaling pathway can promote the cell cycle arrest in non-malignant cells, it serves as a tumor suppressor in the early stage of tumor development. In the advanced stages of cancer, the TGF-β pathway promotes epithelial–mesenchymal transition and produces cytokines, which recruit the immature bone marrow-derived myeloid cells to the tumor microenvironment. Therefore, the TGF-β pathway can induce tumor development and inhibits anti-tumoral immune responses in the advanced stages [12,13,14].

MicroRNAs (miRNAs) are important regulators of the transcription and translation of key regulatory proteins that have many different roles in cancers [15,16]. MiR-214, also referred to as melano-miR, contributes to tumor migration in melanoma [17]. Since miR-214 can target Wnt/β-catenin, it can substantially promote metastasis in patients with melanoma [18,19]. Thus, this miRNA could act as a double-edged sword during the induction of melanoma. Overall, it has been demonstrated that the elimination of miR-214 in melanoma cells could barricade the metastasis and proliferation of this cancer [17,18].

The separation of particular proteins from snake venoms is exceedingly difficult, and the different isolation approaches are often exceedingly costly. Recombinant proteins promote substantial developments in biomedical biotechnology [6,20].

In the previous research, we successfully expressed and produced recombinant Oxus cobra cytotoxin-II. Consequently, we demonstrated the anti-proliferative properties of recombinant cytotoxin-II (rCTII) in the MCF-7 cell line [6]. To the best of our knowledge, it is the first study aimed at identifying the rCTII antibody binding sites and its intracellular anti-proliferative pathways in melanoma.

## 2. Results

### 2.1. The Expression and Characterizing of Protein Interest

Having produced CTII in the recent work [6], as shown in Figure 1, the purified rCTII was resolved by 12% SDS-PAGE and showed a single band of ≈6.6 kDa. Bradford assay revealed that the concentration of rCTII was 600 μg mL^−1^ in *E. coli* SHuffle^®^ T7 Express.

### 2.2. Mapping of IgG-Binding Epitopes

Knowledge of the cytotoxin II epitope characteristics is pivotal for understanding the pathogenicity and toxicity. In this case, antibody-binding regions were identified by Bepipred Linear Epitope Prediction online software. Based on the result, two stretches of amino acids, 6–16 and 19–44, were recognized as B-cell epitopes. The result of epitope mapping is presented in Figure 2 and Table 1.

### 2.3. Cell Viability

Figure 3 displays the viability curves of the cell dose–response for rCTII in SK-MEL-3 and HFF-2 cells. The MTT assay indicated that rCTII inhibits cell proliferation in a dose-dependent manner. In the SK-MEL-3 cancer cell line, the half maximal inhibitory concentration (IC50) value of rCTII was 17.7 µg/mL; however, the IC50 value of rCTII was approximately 24.76 µg/mL in the HFF2 cell line. The higher IC50 values of rCTII in non-tumoral cells compared to melanoma cells are illustrated in Figure 3 (*p* < 0.0001).

### 2.4. The mRNA Expression Levels of Caspase-8 and Caspase-9 and MMP-3 after Treatment by rCTII

To assess the underlying pathways of apoptosis, the expression of caspase-8 and caspase-9 was evaluated by real-time PCR in the control and the treated group to evaluate the underlying pathways of apoptosis.

Following the treatment of cells with rCTII, caspase-8 and caspase-9 were significantly increased in the SK-MEL-3 cells line (both *p* < 0.01) (Figure 4A,B). After treating melanoma cells with rCTII, the ratio of caspase-8/caspase-9 was 2.513, highlighting the dominance of the intrinsic pathway of apoptosis (Figure 5A). Compared to untreated malignant cells, there was a significant down-regulation in the expression level of MMP-3 in treated malignant cells (*p* < 0.05) (Figure 5B).

### 2.5. The mRNA Expression Levels of SMAD-Dependent TGF-β Signaling-Related Genes after Treatment by rCTII

The *SMAD2* and *SMAD3* expression levels were evaluated in the treated and untreated SK-MEL-3 cell line to investigate the rCTII effect on the TGF-β signaling pathway. Following treating melanoma cells with rCTII, there was a significant down-regulation in the expression level of *SMAD2* and *SMAD3* compared to untreated melanoma cells (both *p* < 0.01) (Figure 6A,B).

### 2.6. The Expression Level of miR-214 (melano-miR) after Treating with rCTII

Regarding the pivotal impact of miR-214 on promoting the metastasis of melanoma, the expression of miR-214 was evaluated after treating SK-MEL-3 cells with rCTII. Following treatment of the cancerous cells with rCTII, there was a significant reduction in the expression level of miR-214 compared to untreated melanoma cells (*p* < 0.05) (Figure 6C).

## 3. Discussion

Although there have been considerable advances in treating melanoma patients, this troublesome malignancy has a poor prognosis [3]. Therefore, there is an urgent need to identify novel and strongly effective anticancer medications for patients with melanoma. Recently, cytotoxin-II, the cytotoxin derived from the venom of *Naja naja oxiana*, has shown promising cytotoxicity in breast cancer cells [6,21]. The current study has demonstrated that rCTII might substantially stimulate the apoptosis pathways, especially the intrinsic pathway, inhibit the expression of mir-214 and *MMP-3*, and suppress the TGF-β pathway via down-regulating *SMAD2* and *SMAD3* in melanoma. Of interest, this novel cytotoxin has exerted more cytotoxicity on the melanoma cells compared to non-tumoral cells, indicating its tolerability. This study has also predictively considered two regions of amino acids, i.e., 19–44 and 6–16, as epitopes. As far as we can tell, this is the first study that has predicted the epitope mapping of rCTII and the rCTII-induced signaling pathways in melanoma.

Since cytotoxin-II can stimulate apoptosis via multiple mechanisms such as activating caspases, it has demonstrated potent anticancer properties against tumoral cells [10,21]. Consistent with this, our results have shown that rCTII can significantly inhibit SK-MEL-3 cell proliferation with a lower IC50 compared to HFF-2 cells (17.7 µg/mL and 24.76 µg/mL (*p* < 0.0001), respectively). Since the IC50 value of rCTII in tumoral cells is significantly lower than normal cells, it might bring an ample opportunity for its translation into clinical settings as an acceptable anticancer medication.

The extrinsic and intrinsic pathways, two classic arms of apoptosis, are initiated by caspase-8 and caspase-9, respectively [22]. Dysregulated apoptosis can endow the malignant cells with a selective survival superiority over non-tumoral cells [23]. This current study has revealed that rCTII can remarkably induce apoptosis, mostly via the intrinsic pathway, in melanoma cells. In line with our results, Park et al. have reported that the snake venom toxin from *Vipera lebetina turanica* can suppress neuroblastoma and colorectal cells via stimulating apoptosis [24,25]. In breast cancer, the administration of *Walterinnesia aegyptia* venom has been linked with apoptosis restoration via up-regulating caspase-3, caspase-8, and caspase-9 [26]. Furthermore, the treatment of neuroblastoma cells with the *Naja naja atra* cardiotoxin 3 has increased apoptosis via up-regulating caspase-9 and caspase-3 [27]. These findings, along with our study, have highlighted the potential anticancer property of snake venom toxins, e.g., rCTII, which can trigger apoptosis in tumoral cells.

MMPs, via interactions with the extracellular matrix, play a critical role in promoting tumor migration [28,29]. MMP-3 as one of the most important MMPs has been correlated to metastases by promoting epithelial–mesenchymal transition (EMT), it is also capable of activating MMP-1, which increases the capacity of the tumor cells to invade [30]. We have reported that treating melanoma cells with rCTII can remarkably reduce the expression level of MMP-3, which may have an important role in the suppression of the metastasis. In hepatocellular cancer, the transfection of the snake venom cystatin into tumoral cells has suppressed the tumor invasion and metastasis via suppressing MMP-2, MMP-9, and epithelial–mesenchymal transition [31]. Moreover, this toxin has down-regulated the metastasis of melanoma both in vivo and in vitro [32]. In melanoma cells, the treatment of recombinant BJ46a, a toxin from *Bothrops jararaca* venom, has suppressed MMPs activity and prevents the melanoma cell migration [33]. Furthermore, our previous investigation has indicated that rCTII can robustly decrease *MMP-3* and *MMP-9* expression in breast cancer [6]. Thus, snake venom toxins, e.g., rCTII, can down-regulate *MMP-3* and may suppress metastasis in tumoral cells, especially in melanoma.

SMAD2 and SMAD3 are two members of receptor-regulated SMAD, which have pivotal roles in the TGF-β signaling pathway [34]. In this signaling pathway, TGF-β binds to the TGF-β type II receptor and recruits the type I receptor, which subsequently leads to the phosphorylation of SMAD2/3. The phosphorylated SMAD2/3 forms a complex with SMAD4, which can regulate the transcription of target genes [35]. In the initial stage of tumorigenesis, the TGF-β pathway promotes apoptosis and cell arrest; however, the TGF-β signaling pathway promotes cancer development and metastasis via inducing epithelial–mesenchymal transition, suppressing anti-tumoral immune responses and producing cytokines [14,35,36]. Our study has shown that the treatment of melanoma cells with rCTII can considerably reduce the expression levels of *SMAD2* and *SMAD3* genes, which can inhibit metastasis and tumor development. In line with this, the cytotoxic effect of Akbu-LAAO, a toxin from *Agkistrodon blomhoffii ussurensis*, has been associated with the inhibition of the TGF-β pathway [37]. Therefore, snake toxins, e.g., rCTII, can inhibit metastasis via the TGF-β-SMAD signaling pathway.

Since miR-214 can elevate the expression level of CD166 in melanoma cells, its up-regulation has been associated with the metastasis of melanoma [38]. However, miR-214 can serve as a tumor suppressor in cervical and colorectal cancer [39]. The current study has demonstrated that rCTII can decrease the expression level of miR-214, which may confirm the anti-proliferation feature of this novel cytotoxin.

## 4. Materials and Methods

### 4.1. Bacterial Strain and Cell Culture

SHuffle^®^ T7 Express Competent *E. coli* was considered as the expression host for the production of the rCTII. SK-MEL-3 (malignant melanoma cell line) and HFF-2 (Normal melanoma cell line) were purchased from the National Cell Bank of Iran and then were cultured in RPMI-1640 medium, supplemented with 10% fetal bovine serum (FBS) and penicillin/streptomycin mixtures (Gibco, Carlsbad, CA, USA). The cells were subsequently cultivated in the incubator in a humidified atmosphere at 37 °C and with 5% CO_2_ level.

### 4.2. The Expression and Purification of CTII in SHuffle^®^ T7 Express

The DNA sequence referring to CTII protein (Naja oxiana, UniProtKB: P01441) was codon-optimized for protein expression in *E. coli*. The recombinant pET28a-SUMO-CTII vector, which was developed previously [6], was transformed into competent SHuffle^®^ T7 Express cells. The transformed clones were selected using Luria broth (LB) agar medium (Merk, Germany) containing 50 µg mL^−1^ of kanamycin. Then, a single colony of transformed *E. coli,* transferred to 10 mL of LB liquid medium and was incubated at incubator (overnight shaking 37 °C, 150 rpm). After incubation, 1 mL of this medium was poured into 100 mL of LB medium (50 µg mL^−1^ Kanamycin). Subsequently, this medium was incubated at 37 °C until they reached the exponential phase; then, the expression processes continued through the addition of 1 mM isopropyl-β-D-thiogalactopyranoside (IPTG) and incubated for 4 h at 28 °C. At the end of the expression time, the pellets of induced bacteria were collected (7000× *g* centrifuge at 4 °C for 20 min) and then resuspended in lysis buffer and was sonicated on ice. Then, all debris was collected after 7000× *g* centrifugation at 4 °C for 30 min. The final fusion protein was added to the affinity chromatography column packed with immobilized metal ion-affinity chromatography (IMAC) (Qiagen, Germany) [6].

### 4.3. Cleavage of SUMO Fusions, Quantification of rCTII, and SDS-PAGE

As mentioned in the previous research, the protein of interest was cleaved according to the SUMO Protease manufacturer’s instrument (Thermo Fisher Scientific, San Jose, CA, USA). The rCTII was purified by using the Ni-NTA column. Then, the concentration of rCTII was assessed by the Bradford method. Assessment of the size and purity of the rCTII was evaluated by protein separation in 12% sodium dodecyl sulfate-polyacrylamide gel electrophoresis (SDS-PAGE) and subsequent staining with silver nitrate [6].

### 4.4. The Epitope Mapping of Cytotoxin-II

The prediction of B-cell epitopes has been conducted to discover potential epitopes on the recombinant cytotoxin-II. As the corresponding 3D model of recombinant protein was not found in Protein Data Bank archive (PDB), the ab initio approach was conducted using the I-TASSER server service (https://zhanglab.ccmb.med.umich.edu/I-TASSER/). Therefore, the generation of the 3D model was conducted by collecting the highest scoring of the template structure based on the target model of the CTII amino acid sequence. The I-TASSER developed models were evaluated depending on the confidence score (C-score), which should be in the range of −5 to 2. Linear B-cell epitope prediction was conducted using the BepiPred modified hidden Markov model from the Immune Epitope Database (IEDB) analysis resource v2.12 (http://tools.immuneepitope.org/bcell/) [40]. The recommended cut-off of 0.5 was applied to determine potential B-cell linear epitopes, ensuring a sensibility of 58% and specificity of 57% to this approach. Linear B-cell epitopes are predicted to be located at the residues with the highest scores.

### 4.5. Proliferation Assay

An MTT (3-(4,5-dimethylthiazol-2-yl)-2,5-diphenyltetrazolium bromide) assay was performed to examine the viability of SK-MEL-3 and HFF-2 cells after using rCTII. In brief, cells were cultivated (15,000 cells in each well) in 96-well plates for 24 h at 75% confluency. Cells have been treated with various amounts of recombinant CTII for 24 h. Cells were exposed to MTT (5 mg/mL) at the end of treatment and incubated at 37 °C for 4 h. After removal of the supernatant, 100 μL of DMSO was applied to each well, and the plate was placed in the incubator for ten minutes to dissolve the crystals of formazan. Absorbance for each plate well was assessed using a microplate reader (Tecan, Mannedorf, Switzerland) at a test wavelength of 570 nm and a reference wavelength of 690 nm. The optical density (OD) was measured as the absorbance of the reference wavelength minus that of the test wavelength.

### 4.6. RNA Extraction, cDNA Synthesis, and qRT-PCR

After the treatment of the SK-MEL-3 cancer cell line with rCTII (IC50), the total RNA extraction was conducted by the TRIzol reagent (RiboEx) following the manufacturer’s protocol. The complementary DNA (cDNA) was synthesized using 1 μg of total RNA. The qRT-PCR reactions were conducted in the light cycler 96 system (Roche, Germany) to identify transcript levels of target genes. The expression of caspase-8, caspase-9, matrix metallopeptidase 3 (*MMP-3*), *SMAD2, SMAD3*, and miR-214 was analyzed by qRT-PCR. 18S ribosomal RNA was used as the internal control gene. The primer sequences are listed in Table 2. The gene expression ratio of each group was estimated with the 2^−ΔΔCt^ method. All reactions were conducted in triplicate to confirm reproducibility.

### 4.7. Statistical Analysis

The statistical significance of differences between variables was measured by *T*-test analyses by using “pcr” [41] and ggplot2 [42] packages of R software v4.0.2. MTT analysis was performed by using GraphPad Prism 6 software (San Diego, CA, USA). The *p*-values below 0.05 were considered relevant.

## 5. Conclusions

To the best of our knowledge, this study has introduced rCTII as a potent anti-proliferative effect on melanoma cells for the first time. This toxin has substantially suppressed the TGF-β–SMAD signaling pathway and increased apoptosis, mainly via the intrinsic pathway. Furthermore, rCTII has down-regulated the expression levels of miR-214 and *MMP-3*, which may inhibit metastasis in melanoma. This study has also predicted two regions of amino acids, i.e., 19–44 and 6–16, as B-cell epitopes, the understanding of these amino acids is essential for the toxicity of this novel cytotoxin. More studies, in particular on animal models or through clinical trials, are needed to prove our findings.

## Figures and Tables

**Figure 1 molecules-25-05148-f001:**
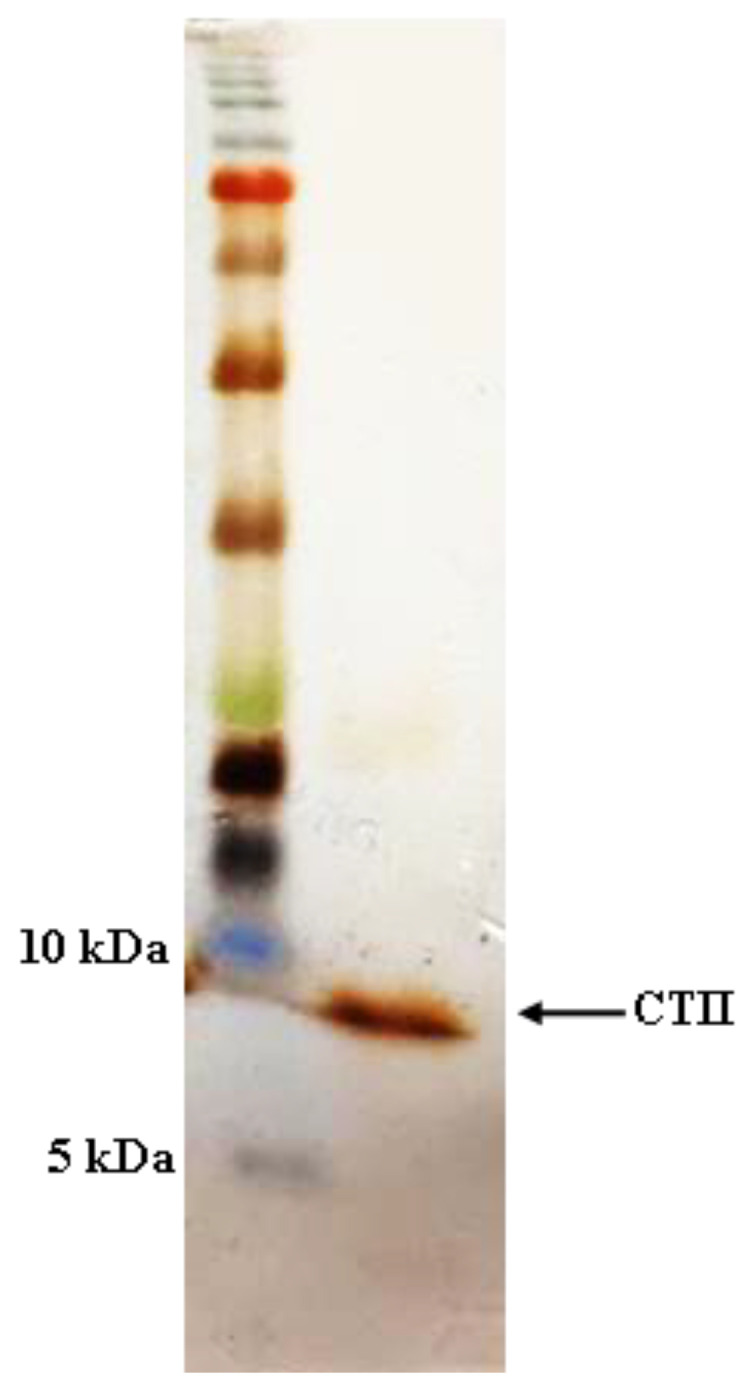
SDS-PAGE analysis of expressed and purified recombinant cytotoxin II.

**Figure 2 molecules-25-05148-f002:**
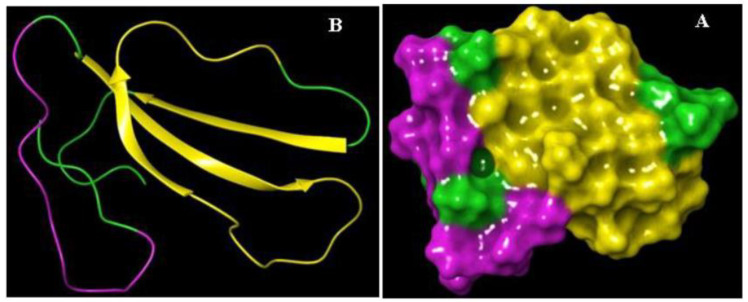
The schematic representation of the 3D structure and epitopes of recombinant cytotoxin-II (rCTII). (**A**) Cartoon and (**B**) surface models.

**Figure 3 molecules-25-05148-f003:**
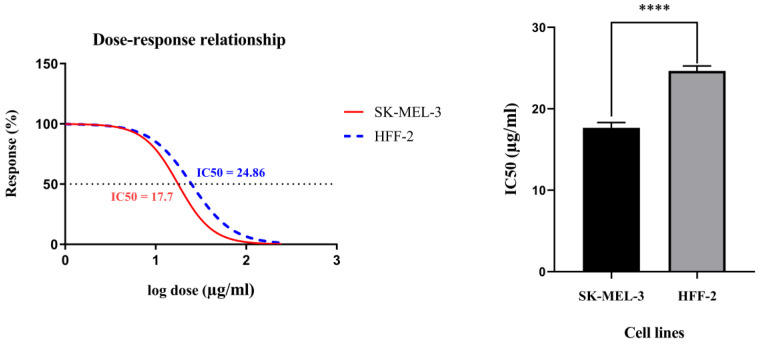
The viability curves of the cell dose–response of recombinant cytotoxin-II in HFF-2 and SK-MEL-3 cell lines. The cells were treated with different concentrations of recombinant cytotoxin-II for 24 h, and at the end of the incubation time, cell viability was determined by the MTT (3-(4,5-dimethylthiazol-2-yl)-2,5-diphenyltetrazolium bromide) reduction assay. The SK-MEL-3 cell line (Red) and HFF2 cell line (Blue). Error bars display the ± SD. The independent *t*-tests assessed the *p*-values. *p* < 0.0001 (****).

**Figure 4 molecules-25-05148-f004:**
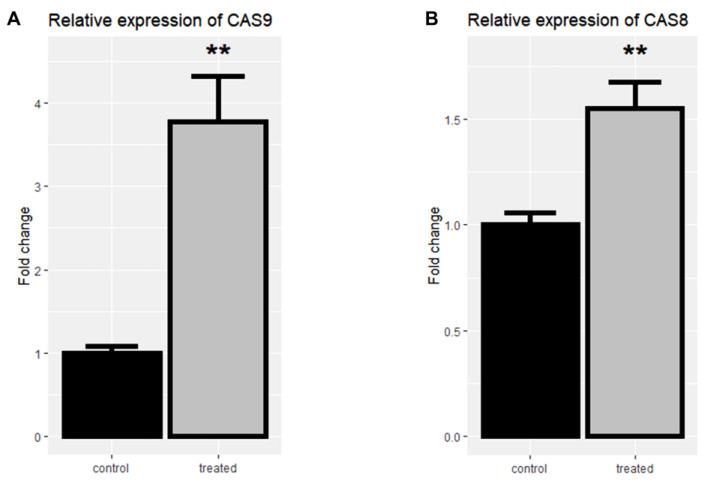
The relative mRNA expression of *Casp-9* (**A**) and *Casp-8* (**B**) in the SK-MEL3 cell line after treatment with 17.7 µg/mL rCTII. The expression of candidate genes was assessed by real-time PCR in the control and treated group. Our result illustrated that this protein potentially activated both intrinsic and extrinsic pathways related to apoptosis. Error bars display the ± SD. The independent *t*-tests assessed the *p*-values. *p* < 0.01 (**).

**Figure 5 molecules-25-05148-f005:**
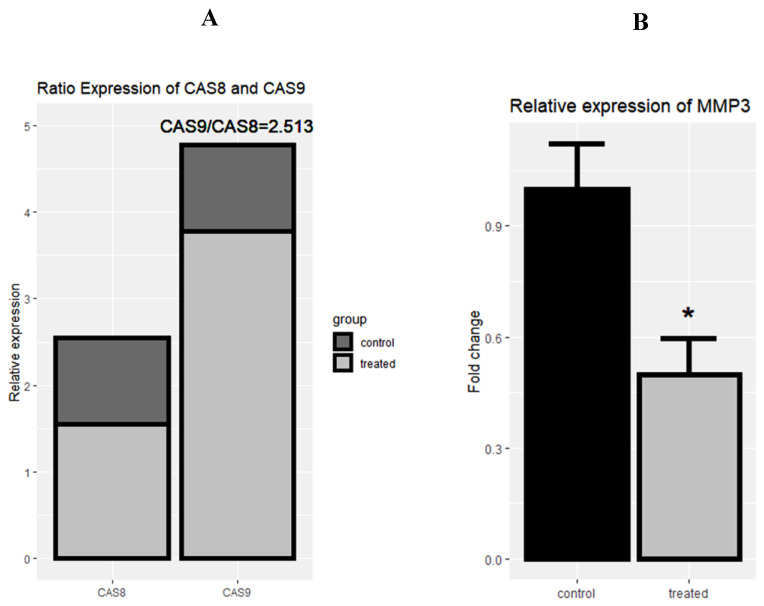
The rational mRNA expression of *Casp-8* and *Casp-9* after the treatment with 17.7 µg/mL rCTII in the SK-MEL-3 cell line (**A**). The relative expression of matrix metallopeptidase 3 (MMP-3) after the treatment with rCTII (**B**). The expression of candidate genes was evaluated through real-time PCR in the control and treated group. Our result illustrated that this protein potentially activated the intrinsic pathway of apoptosis more than extrinsic pathways. In addition, there was a significant down-regulation in the expression level of *MMP-3* in treated cancerous cells. Error bars display the ± SD. The independent *t*-tests assessed the *p*-values. *p* < 0.05 (*).

**Figure 6 molecules-25-05148-f006:**
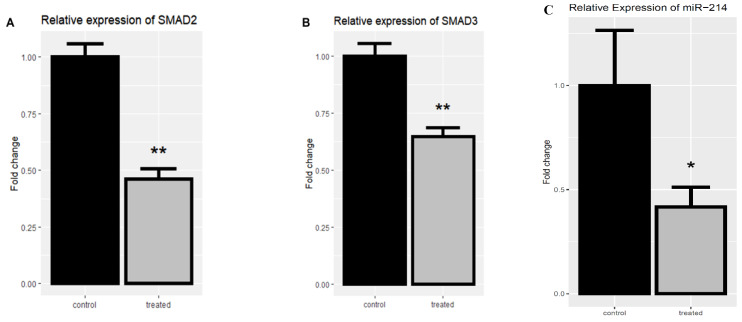
The relative mRNA expression of *SMAD2* and *SMAD3* after the treatment with 17.7 µg/mL rCTII in the SK-MEL-3 cell line (**A**,**B**). The expression level of miR-214 after the treatment with rCTII (**C**). The expression of candidate genes was assessed through real-time PCR in the control and treated group. Our result demonstrated that for the rCTII protein, the *SMAD2* and *SMAD3* expression levels were evaluated in the treated and untreated cell line to investigate the rCTII effect on the transforming growth factor-beta (TGF-β) signaling pathway. After melanoma cells were treated with rCTII, there was a significant down-regulation in the expression level of *SMAD2* and *SMAD3* compared to untreated melanoma cells (both *p* < 0.01). Subsequent to cancerous cells being treated with rCTII, there was a significant reduction in the expression level of miR-214 compared to untreated melanoma cells (*p* < 0.05). Error bars display the ± SD. The independent *t*-tests assessed the *p*-values. *p* < 0.05 (*), *p* < 0.01 (**).

**Table 1 molecules-25-05148-t001:** B-cell epitope prediction for CTII.

Method	Region	Residues	Length	Color	Epitope
Bepipred Linear Epitope Prediction	1–5	LKCKK	5	Green	-
	6–16	LVPLFYKTCPA	10	Purple	yes
	17–18	GK	2	Green	-
	19–44	NLCYKMFMVSNLTVPVKRGCIDVCPK	26	Yellow	yes
	45–60	SSLLVKYVCCNTDKCN	15	Green	-

**Table 2 molecules-25-05148-t002:** The sequence of primers was used in the current study.

Primers	Sequences (5′→3′)
Caspase-8	CTGGTCTGAAGGCTGGTTGTT GTGACCAACTCAAGGGCTCAG
Caspase-9	CCTTCCTCTCTTCATCTCCTGCT TTGCTGTGAGTCCCATTGGT
SMAD2	AAGGGTGGGGAGCAGAATAC CTTGAGCAACGCACTGAAGG
SMAD3	ACTACATCGGAGGGGAGGTC GGGTCAACTGGTAGACAGCC
MMP-3	TACTGGAGATTTGATGAGAAGAG TACAGATTCACGCTCAAGTTCC
18S rRNA	CTACCACATCCAAGGAAGGCA TTTTTCGTACTACCTCCCCG
miR-214	AACAAGACAGCAGGCACAGA GTCGTATCCAGTGCAGGGT SL: GTCGTATCCAGTGCAGGGTCCGAGGTATTCGCACTGGATACGACACTGCC

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
