# Peer review of "Targeting TGF-β-Mediated SMAD Signaling Pathway via Novel Recombinant Cytotoxin II: A Potent Protein from Naja naja oxiana Venom in Melanoma"

_molecules, 2020, doi:10.3390/molecules25215148_

Round 1
Reviewer 1 Report
This study has introduced rCTII as a potent antiproliferative effect on melanoma cells. This venom snake toxin has substantially suppressed the TGF253 β-SMAD signaling pathway and increased apoptosis, mainly via the intrinsic pathway. Furthermore, rCTII has down-regulated the expression levels of miR-214 and MMP-3, which can inhibit metastasis in melanoma. This study has also highlighted two regions of amino acids, i.e., 19-44 and 6-16, as B256 cell epitopes, which their understanding is essential for the toxicity of this novel cytotoxin. The low IC50 value of rCTII in melanoma compared to non-tumoral cells indicates the tolerability of this potent cytotoxin.
It is an original work and with an adequate scientific content. the study opens the possibility of a possible treatment of melanomas. Therefore, I consider that it can be accepted, after correcting some minor errors in English.
Author Response
This study has introduced rCTII as a potent antiproliferative effect on melanoma cells. This venom snake toxin has substantially suppressed the TGF253 β-SMAD signaling pathway and increased apoptosis, mainly via the intrinsic pathway. Furthermore, rCTII has down-regulated the expression levels of miR-214 and MMP-3, which can inhibit metastasis in melanoma. This study has also highlighted two regions of amino acids, i.e., 19-44 and 6-16, as B256 cell epitopes, which their understanding is essential for the toxicity of this novel cytotoxin. The low IC50 value of rCTII in melanoma compared to non-tumoral cells indicates the tolerability of this potent cytotoxin.
It is an original work and with an adequate scientific content. the study opens the possibility of a possible treatment of melanomas. Therefore, I consider that it can be accepted, after correcting some minor errors in English.
- We appreciate your comment and apologize for this error. The English structure and grammar of the manuscript have been thoroughly reviewed and several modifications to the initial manuscript have been made.
Reviewer 2 Report
This paper is interesting and it contains valuable and somehow promising data on important proteins related to cell signaling, proliferation and apoptosis. Expression of some biochemical markers are studied.
However, before publication, the paper would need significant improvements concerning the following points:
Lines 45-46: Letter size should be unform, but line 45 is not. References should be properly used. I.e. ref, 3 is not enough to justify the outcome of melanoma. Look for others with a general landscape about melanoma. Ref. 20 in this paper is mor appropriate that Ref. 3
Introduction, Methods (2.2. The expression and purification of CTII in SHuffle® T7 Express) and Results ( 3.1. The expression and characterizing of protein interest) are greatly based on Reference 5.
So, this ref 5 is a cornerstone reference, as it is cited several times but it is not possible to consult. Is the reference incomplete? Journal? Doi?. . At the current form, the ref. is not accessible for reviewer and future readers.
Some data supposedly contained in this reference should be included in this paper. More details should be added, especially to the sections 2.2 and 3.1.
Figure 3: Please, plot the 2 curves in the same figure to better observe the small difference in the sensitivity of the two cell lines to cytotoxin-II.
Figure 4: This is relative expression of mRNA, not protein, and this should be clearly indicated. Please label at least Casp (or caspase) to avoid confusion with Cas nucleases associated to CRISPR.
Same for Figures 5, 6
Figure legends are poor. More precise details are needed. Conditions of treatment and so on.
Indicate something else and how the two epitopes have been characterized. This is just a theoretical simple proposal. To state that this research is the first study to identify those regions is a little bit exaggerated
Differences in IC50 between malignant and normal melanocytes is not significant yet to discuss that as a tolerable anticancer medication. Please, re-write lines 202-207.
Related to that, please, delete or modify last sentence in conclusion (lines 256-258). Lines 246-249 could be translated to conclusion, but "can robustly" should change by other lighter expression such as "could be used". The current conclusion to exaggerated, too resounding according available data. Expectations should be decreased.
About MMPs (lines 219-228), this study is only referred to MMP3. The reasons why this is the only MMP determined would be discussed.
English would be edited to correct some grammatical errors. Remove merged words throughout the text
Author Response
This paper is interesting and it contains valuable and somehow promising data on important proteins related to cell signaling, proliferation and apoptosis. Expression of some biochemical markers are studied. However, before publication, the paper would need significant improvements concerning the following points:
Lines 45-46: Letter size should be unform, but line 45 is not. References should be properly used. I.e. ref, 3 is not enough to justify the outcome of melanoma. Look for others with a general landscape about melanoma. Ref. 20 in this paper is mor appropriate that Ref. 3
- Thanks for the mentioned point. As suggested, the whole structure of the manuscript checked, and also, all the references have been reviewed and cited correctly.
Introduction, Methods (2.2. The expression and purification of CTII in SHuffle® T7 Express) and Results ( 3.1. The expression and characterizing of protein interest) are greatly based on Reference 5. So, this ref 5 is a cornerstone reference, as it is cited several times but it is not possible to consult. Is the reference incomplete? Journal? Doi?. . At the current form, the ref. is not accessible for reviewer and future readers. Some data supposedly contained in this reference should be included in this paper. More details should be added, especially to the sections 2.2 and 3.1.
- We appreciate your suggestion. We have added more details about the characterizing, expression, and production of rCTII which we previously published in the International Journal of Biological Macromolecules. Also, we reviewed and edited all the references in the current version.
Figure 3: Please, plot the 2 curves in the same figure to better observe the small difference in the sensitivity of the two cell lines to cytotoxin-II.
- Thank you for your suggestion. In the new version of the manuscript, we provided a plot that can show the difference in a great way as well as showed the significance of IC50 in both cell lines (bar chart).
Figure 4: This is relative expression of mRNA, not protein, and this should be clearly indicated. Please label at least Casp (or caspase) to avoid confusion with Cas nucleases associated to CRISPR. Same for Figures 5, 6. Figure legends are poor. More precise details are needed. Conditions of treatment and so on.
- We apologize for the mistake. It has been corrected in the revised manuscript and more details have added to the figure legends (highlighted).
Indicate something else and how the two epitopes have been characterized. This is just a theoretical simple proposal. To state that this research is the first study to identify those regions is a little bit exaggerated
- We edited and rephrased the mentioned paragraphs
Differences in IC50 between malignant and normal melanocytes is not significant yet to discuss that as a tolerable anticancer medication. Please, re-write lines 202-207.
- Thanks to the point, we used statistical software to analyses the difference between two IC50 in both cell lines, As mentioned in the current version, we showed there is a significant difference between two of them. (Please see Fig3)
Related to that, please, delete or modify last sentence in conclusion (lines 256-258). Lines 246-249 could be translated to conclusion, but "can robustly" should change by other lighter expression such as "could be used". The current conclusion to exaggerated, too resounding according available data. Expectations should be decreased.
- Thanks for pointing out the issue. We have deleted and modified as you mentioned.
About MMPs (lines 219-228), this study is only referred to MMP3. The reasons why this is the only MMP determined would be discussed.
- Thanks for your careful reading of our manuscript. We have added as you mentioned. MMPs, via interactions with the extracellular matrix, play a critical role in promoting tumor migration. MMP3 as one of the most important MMPs has been correlated to metastases by promoting Epithelial-mesenchymal transition (EMT), it also, capable to activate MMP-1, which increases the capacity of the tumor cells to invade. Here we tried to evaluate one of the essential MMPs, MMP3, which is overexpressed in Melanoma. Please see this ref[1]
English would be edited to correct some grammatical errors. Remove merged words throughout the text
- We appreciate your comment and apologize for this error. The English structure and grammar of the manuscript have been thoroughly reviewed and several modifications to the initial manuscript have been made.
Reference:
- Bastian, A.; Nichita, L.; Zurac, S. Matrix Metalloproteinases in Melanoma with and without Regression. The Role of Matrix Metalloproteinase in Human Body Pathologies 2017, 145.
Round 2
Reviewer 2 Report
The authors have fulfilled the concerned points satisfactorily: the significance of the difference between the IC50 in both cell types is something to be further investigated.